# Behind the Myth of Exploration in Policy Gradients

## Abstract

In order to compute near-optimal policies with policy-gradient algorithms, it is common in practice to include intrinsic exploration terms in the learning objective. Although the effectiveness of these terms is usually justified by an intrinsic need to explore environments, we propose a novel analysis with the lens of numerical optimization. Two criteria are introduced on the learning objective and two others on its stochastic gradient estimates, and are afterwards used to discuss the quality of the policy after optimization. The analysis sheds light on two separate effects of exploration techniques. First, they make it possible to smooth the learning objective and to eliminate local optima while preserving the global maximum. Second, they modify the gradient estimates, increasing the probability that the stochastic parameter updates eventually provide an optimal policy. We empirically illustrate these effects with exploration strategies based on entropy bonuses, identifying limitations and suggesting directions for future work.

## 1  Introduction

Reinforcement learning (RL) has been successful in complex decision-making problems, including playing games (Mnih et al., 2015; Silver et al., 2017), operating power systems (Aittahar et al., 2024), controlling robots (Kalashnikov et al., 2018), and trading electricity (Boukas et al., 2021).

Reinforcement learning algorithms interact with an environment to gather information about this environment, which in turn enables to compute and follow an optimal policy. This interaction creates a trade-off between exploration and exploitation. In short, to eventually compute a good policy, an agent must obtain additional information about the environment by taking actions that are likely not optimal. In algorithms where this trade-off is explicit, exploration is well understood and has been the subject of many works (Dann et al., 2017; Azar et al., 2017; Neu & Pike-Burke, 2020). In policy-gradient algorithms, one often can not distinguish exploration from exploitation. Nevertheless, a key theoretical requirement for convergence to globally (or even locally) optimal solutions is that policies remain sufficiently stochastic during the learning procedure (Bhandari & Russo, 2019; Bhatt et al., 2019; Agarwal et al., 2020; Zhang et al., 2021a; Bedi et al., 2022). Interestingly, neither softmax nor Gaussian policies guarantee enough stochasticity to ensure (fast) convergence (Mei et al., 2020; 2021; Bedi et al., 2022). This requirement for stochasticity in policy gradients is often loosely called exploration and understood as the need to infinitely sample each state-action pair.

The need for sufficient policy randomness has long been known by practitioners, who have attempted to meet this requirement in policy-gradient methods with reward shaping. In these approaches, a learning objective that promotes or hinders behaviors by assigning reward bonuses to some states and actions is optimized as a surrogate for the policy return. Such bonuses typically encourage actions that reduce uncertainty of the agent about its environment (Pathak et al., 2017; Burda et al., 2018; Zhang et al., 2021c) or that maximize the entropy of states or actions (Williams & Peng, 1991; Bellemare et al., 2016; Haarnoja et al., 2019; Hazan et al., 2019; Lee et al., 2019; Islam et al., 2019; Guo et al., 2021; Zhang et al., 2021b).

Optimizing surrogate learning objectives that include reward bonuses appears particularly effective for solving challenging tasks, typically those with complex dynamics or sparse rewards. The connection between such objectives and exploration has nevertheless long been limited to intuition, and the lack of theoretical analysis has led to common folklore that seeks to justify their practical efficiency. Research on these objectives has

mainly focused on entropy regularization (an exploration-based reward shaping technique in which the policy entropy is added to the learning objective). This line of work includes the empirical study of Ahmed et al. (2019), which concludes that entropy regularization helps provide smoother objective functions. The same strategy has been reinterpreted as a robust optimization method by Husain et al. (2021) and, equivalently, as a two-player game by Brekelmans et al. (2022). Bolland et al. (2023) further argue that optimizing an entropy-regularized objective is equivalent to maximizing the return of another policy with larger variance. Finally, convergence rates for policy-gradient algorithms with entropy regularization have recently been derived (Cen et al., 2022; Zhang et al., 2021a). More general studies of learning dynamics have examined the influence of baselines in policy-gradient methods (Chung et al., 2021) and potential-based reward shaping strategies that do not modify the learning objective (Ng et al., 1999; Wiewiora et al., 2003; Harutyunyan et al., 2015; Forbes et al., 2024). All of these studies remain restricted to particular learning objectives and the literature still lacks unified explanations of the improvements intrinsic reward bonuses can provide, the conditions under which they arise, and systematic ways to compare different bonuses, among others. This work aims to take a step toward answering these questions.

Before delving into our contributions, we recall that the convergence of stochastic ascent methods is driven by the objective function and how the ascent directions are estimated. First, the objective function shall be (pseudo) concave to find its global maximum (Bottou, 1998). Second, the convergence rate is influenced by the distribution of the stochastic ascent estimates (Chen & Luss, 2018; Ajalloeian & Stich, 2020). In this paper, we rigorously study policy-gradient methods with exploration-based reward shaping through the lens of these two optimization theory aspects. To that end, we first introduce two new criteria that relate the return of a policy to the learning objective with exploration bonuses, and their respective optima. Second, we introduce two additional criteria on the distribution of the gradient estimates of the learning objective and their likelihood of providing directions in which the learning objective and the return increase. Importantly, these criteria are general to any reward-shaping strategy, and highlight the importance of reward shaping that modify the optimal control behavior, in opposition to the literature on potential-based reward shaping. The influence of some exploration bonuses are illustrated and discussed in the light of these four criteria. In practice, finding good exploration strategies is problem specific and we thus introduce a general framework for the study and interpretation of exploration in policy-gradient methods instead of trying to find the best exploration method for a given task.

The paper is organized as follows. In Section 2, we provide the background about policy gradients and exploration. Section 3 focuses on the effect of exploration on the learning objective while Section 4 is dedicated to the effect on the gradient estimates used in the policy-gradient algorithms. Both sections include illustrations on pathological examples, and Section 5 presents extended experiments[1]. Conclusions and future works are discussed in Section 6.

## 2 Background

Let us introduce Markov decision processes and policy gradients with intrinsic exploration.

### 2.1 Markov Decision Processes

We study problems in which an agent makes sequential decisions in a stochastic environment (Sutton & Barto, 2018). The environment is modeled as an infinite-time Markov decision process (MDP) composed of a state space $\mathcal{S}$, an action space $\mathcal{A}$, an initial state distribution with density $p_0$, a transition distribution (modeling the dynamics) with conditional density $p$, a bounded reward function $\rho$, and a discount factor $\gamma \in [0, 1)$. When an agent interacts with the MDP, first an initial state $s_0 \sim p_0(\cdot)$ is sampled, then, at each time step $t$, the agent provides an action $a_t \in \mathcal{A}$, leading to a new state $s_{t+1} \sim p(\cdot|s_t, a_t)$. Finally, a reward $r_t = \rho(s_t, a_t) \in \mathbb{R}$ is observed after each action $a_t$.

A policy $\pi \in \Pi = \mathcal{S} \to \mathcal{P}(\mathcal{A})$ is a mapping from the state space $\mathcal{S}$ to the set of probability measures on the action space $\mathcal{P}(\mathcal{A})$, where $\pi(a|s)$ is the associated conditional probability of action $a$ in state $s$. The function $J : \Pi \to \mathbb{R}$ is defined as a function mapping any policy $\pi$ to the expected discounted sum of rewards gathered

---

[1]Experimental details and implementations can be found at `https://github.com/anonymized`.

by an agent interacting with the MDP by sampling actions from $\pi$. We call the return of the policy $\pi$ the value provided by that function

$$J(\pi) = \frac{1}{1-\gamma} \mathop{\mathbb{E}}_{\substack{s \sim d^{\pi,\gamma}(\cdot) \\ a \sim \pi(\cdot|s)}} [\rho(s,a)] \ , \tag{1}$$

where $d^{\pi,\gamma}(s) = (1-\gamma) \sum_{\Delta=0}^{\infty} \gamma^\Delta p_\Delta^\pi(s)$ is the state-visitation measure $\forall s \in \mathcal{S}$, and $p_\Delta^\pi$ is the state probability after $\Delta$ time steps following the policy $\pi$. In reinforcement learning, we seek to find an optimal policy $\pi^* \in \Pi$ maximizing the expected discounted sum of rewards $J$.

## 2.2 Policy-Gradient Algorithms

Policy-gradient algorithms (locally) optimize a parameterized policy $\pi_\theta$ to find the parameter $\theta^*$ that maximizes the return of the policy. Naively maximizing the return may provide suboptimal results. This problem is mitigated in practice by exploration strategies, which typically consist of optimizing surrogate learning objectives that encourage certain behaviors. In this work, we consider reward-shaping strategies in which the expected discounted sum of rewards is extended by $K$ additional reward terms $\rho_i^{int}$, called intrinsic motivation terms, and we optimize the learning objective

$$L(\theta) = \frac{1}{1-\gamma} \mathop{\mathbb{E}}_{\substack{s \sim d^{\pi_\theta,\gamma}(\cdot) \\ a \sim \pi_\theta(\cdot|s)}} \left[ \rho(s,a) + \sum_{i=0}^{K-1} \lambda_i \rho_i^{int}(s,a) \right] = J(\pi_\theta) + J^{int}(\pi_\theta) \ , \tag{2}$$

where $\lambda_i$ are non-negative weights for each intrinsic reward, and $J^{int}(\pi_\theta)$ is the intrinsic return of the policy. The parameter maximizing the learning objective is denoted by $\theta^\dagger$, which we distinguish from the optimal policy parameter $\theta^*$. Most of the intrinsic motivation terms can be classified into the following two groups.

**Uncertainty-based motivations.** It is common to provide bonuses for performing actions that reduce the agent uncertainty about an internal environment model (Pathak et al., 2017; Burda et al., 2018; Zhang et al., 2021c). The intrinsic motivation terms are then proportional to the prediction errors of that model, which is typically learned.

**Entropy-based motivations.** It is also common to motivate agents to visit states and/or select actions that are less likely. It is typically achieved using intrinsic rewards that depend on the policy (Williams & Peng, 1991; Haarnoja et al., 2019) or on the state-visitation measure (Bellemare et al., 2016; Haarnoja et al., 2019; Hazan et al., 2019; Lee et al., 2019; Islam et al., 2019)

$$\rho^a(s,a) = -\log \pi_\theta(a|s) \qquad\qquad J^a(\pi_\theta) \propto \mathbb{E}_{s \sim d^{\pi_\theta,\gamma}(\cdot)} \left[ \mathcal{H}_a \left( \pi_\theta(a|s) \right) \right] \tag{3}$$

$$\rho^s(s,a) = -\log d^{\pi_\theta,\gamma}(z) \qquad\qquad J^s(\pi_\theta) \propto \mathcal{H}_z \left( d^{\pi_\theta,\gamma}(z) \right) \ , \tag{4}$$

where $z = \phi(s)$ is a feature built from the state $s$, and where $\mathcal{H}_x(q(x))$ is the entropy of $q$. The corresponding intrinsic returns, $J^a(\pi_\theta)$ and $J^s(\pi_\theta)$, are maximized for policies with actions uniformly distributed in each state, and for policies that visit every feature uniformly, respectively. Importantly, these rewards require estimating the distribution over the states and/or actions, which implicitly depends on the policy parameter $\theta$. Notice that the first technique is usually referred to as entropy regularization.

In this work, we consider on-policy policy-gradient algorithms, which were, among others, reviewed by Duan et al. (2016) and Andrychowicz et al. (2020). These algorithms optimize differentiable parameterized policies with gradient-based local optimization. They iteratively approximate an ascent direction $\hat{d}$, relying on samples from the policy in the MDP, and update the parameters in that ascent direction, or in a combination of previous ascent directions (Hinton et al., 2012; Kingma & Ba, 2014). For the sake of simplicity and without loss of generality, we consider that the ascent direction $\hat{d}$ is the sum of an estimate of the gradient of the return $\hat{g} \approx \nabla_\theta J(\pi_\theta)$ and an estimate of the gradient of the intrinsic return $\hat{i} \approx \nabla_\theta J^{int}(\pi_\theta)$. In practice, the first may be unbiased, while the second is computed by neglecting some partial derivatives of $\theta$ and is thus biased, typically neglecting the influence of the policy on the intrinsic reward.

# 3 Study of the Learning Objective

In this section, we study the influence of the exploration terms on the learning objective defined in equation (2). We define two criteria under which the learning objective can be globally optimized by ascent methods and under which the solution is close to an optimal policy. We then graphically illustrate how exploration modifies the learning objective to remove local extrema.

## 3.1 Policy-Gradient Learning Objective

Policy-gradient algorithms using exploration maximize the learning objective function $L$ as defined in equation (2). We introduce two criteria related to this learning objective to study the performance of the policy-gradient algorithm. First, we say that a learning objective $L$ is $\epsilon$-coherent when its global maximum is in an $\epsilon$-neighborhood of the return of an optimal policy. Second, we call learning objectives that have a unique maximum and no other stationary point pseudoconcave.

**Coherence criterion.** *A learning objective $L$ is $\epsilon$-coherent if, and only if,*

$$J(\pi_{\theta^*}) - J(\pi_{\theta^\dagger}) \leq \epsilon \,, \tag{5}$$

*where $\theta^* \in \arg\max_\theta J(\pi_\theta)$ and where $\theta^\dagger \in \arg\max_\theta L(\theta)$.*

**Pseudoconcavity criterion.** *A learning objective $L$ is pseudoconcave if, and only if,*

$$\exists! \, \theta^\dagger : \nabla L(\theta^\dagger) = 0 \wedge L(\theta^\dagger) = \max_\theta L(\theta) \,. \tag{6}$$

If the pseudoconcavity criterion is satisfied, there is a single optimum, and it is thus possible to globally optimize the learning objective function by (stochastic) gradient ascent (Bottou, 2010)[2]. If the learning objective is furthermore $\epsilon$-coherent, the latter solution is also a near-optimal policy, where $\epsilon$ is the bound on the suboptimality of its return.

Let us finally recall a theorem of Ng et al. (1999).

**Consistency Theorem.** *The learning objective $L$ is $\epsilon$-coherent, with $\epsilon = 0$, in any MDP with state space $\mathcal{S}$, action space $\mathcal{A}$ and discount factor $\gamma$, if, and only if, $J(\pi_\theta) = L(\theta)$ for all $\theta$. Furthermore, the intrinsic rewards are potential-based.*

This theorem establishes a trade-off between the coherence and pseudoconcavity criteria. Either an exploration method guarantees coherence with $\epsilon = 0$ in any MDP, but then the learning objective equals the return and the pseudoconcavity criterion is violated when the return is not pseudoconcave itself. Otherwise, the pseudoconcavity criterion is satisfied, but then the exploration method is MDP-dependent or coherence is guaranteed at best with $\epsilon > 0$.

## 3.2 Illustration of the Effect of Exploration on the Learning Objective

Exploration is of paramount importance in environments with complex dynamics and reward functions, where locally optimal policies may exist (Lee et al., 2019; Liu & Abbeel, 2021; Zhang et al., 2021b). In the following, we first define such an environment and a policy parameterization that will serve as an example to illustrate the effect of exploration on the optimization process. For the sake of the analysis, we then represent the learning objectives associated with different exploration strategies, and depict their global and local optima. Learning objectives with a single optimum satisfy the pseudoconcavity criterion. In addition, we represent the neighborhood $\Omega$ of the optimal policy parameters such that any learning objective with its global maximum within this region is coherent for a given $\epsilon$. In light of the coherence and the pseudoconcavity criteria, we finally elaborate on the policy parameters computed by stochastic gradient ascent algorithms.

---

[2]For the sake of keeping discussions simple, the definition of pseudoconcavity is simplified (Mangasarian, 1975), and additional assumptions on the stochastic gradient estimates are neglected. Furthermore, the literature studying policy-gradient algorithms usually rely on the more complex Polyak-Lojasiewicz inequality (gradient dominance) to guarantee convergence of gradient ascent instead of the pseudoconcavity.

We consider the environment illustrated in Figure 1a, where a car moves in a valley (Bolland et al., 2023). We denote by $x$ and $v$ the position and speed of the car, which together compose its state $s = (x, v)$. The valley contains two valley floors, located at $x_{initial} = -3$ and $x_{target} = 3$, separated by a peak. The car starts at rest $v_0 = 0$ at the higher floor $x_0 = x_{initial}$ and receives rewards proportional to the depth of the valley at its current position. The reward function is shown in Figure 1b. We consider a policy $\pi_{K,\sigma}(a|s) = \mathcal{N}(a|\mu_K(s), \sigma)$ with $\mu_K(s) = K \times (x - x_{target})$, parameterized by the vector $\theta = (K, \sigma)$. It corresponds to a noisy proportional controller. Figure 1c illustrates the contour map of the return, annotated with local optima, and shows in green the set of near-optimal policies. The optimal policy drives the car to reach the lowest floor in $x_{target}$.

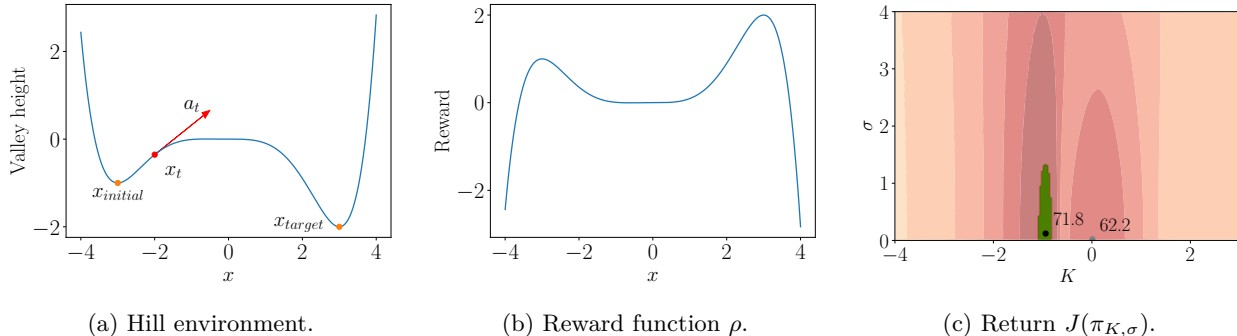

(a) Hill environment.  (b) Reward function $\rho$.  (c) Return $J(\pi_{K,\sigma})$.

Figure 1: Illustration of the *hill environment* in Figure 1a and its reward function in Figure 1b. In Figure 1c, the return of the policy $\pi_{K,\sigma}$ with the global and local maximum represented in black and grey, together with their respective return values. In green $\Omega = \{\theta' | \max_\theta J(\pi_\theta) - J(\pi_{\theta'}) \leq \epsilon = 1\}$.

Figure 2 illustrates the learning objective with the intrinsic rewards $\rho^s(s, a) = -\log d^{\pi_{K,\sigma},\gamma}(\phi(s))$, from equation (4), and $\rho^a(s, a) = -\log \pi_{K,\sigma}(a|s)$, from equation (3), for different values of the corresponding weights $\lambda_s$ and $\lambda_a$. Here, the feature is the position in the valley $\phi(s) = x$. On the one hand, the smaller the exploration weights, the closer the parameter $\theta^\dagger$ maximizing the learning objective is to $\theta^*$, and the smaller $\epsilon$ for which coherence is guaranteed. Furthermore, the return of the policy corresponding to $\theta^\dagger$ is then also higher. In the particular case depicted in the figure, we observe that, only for the smallest weights, the parameter $\theta^\dagger$ maximizing the learning objective, represented by a black dot, is inside the green set $\Omega$, such that $\epsilon$-coherence is guaranteed for $\epsilon = 1$. On the other hand, as the weights increase, we observe that the learning objective eventually becomes pseudoconcave. As explained, this comes at the cost of a higher $\epsilon$ for guaranteeing $\epsilon$-coherence; there is thus a trade-off between the two criteria. In Figure 2b, we observe that in this environment, there is a learning objective that satisfies both the pseudoconcavity criterion and the $\epsilon$-coherence criterion for $\epsilon = 1$. Indeed, there is a single global maximum represented by a black dot, which is part of the set $\Omega$.

Shaping the reward function with an exploration strategy based on the state-visitation entropy appears to be a good solution for optimizing the policy. However, a notable drawback is that the reward depends on the policy, and its (gradient) computation requires estimating a complex probability measure. In this example, the intrinsic reward function itself was estimated by Monte Carlo sampling for each parameter, which would not scale for complex problems and requires approximations and costly evaluation strategies. In Section 5, we extend the study to more complex environments, where the policy is a neural network and the state-visitation probability is approximated.

The observations suggest that well-chosen exploration strategies can lead to learning objective functions that satisfy the two criteria defined in the previous section, thereby guaranteeing that policies that are suboptimal by at most $\epsilon$ can be computed by local optimization. When designing exploration strategies, it is essential to keep in mind that we modify the learning objective so that the algorithms can converge to optimal policy parameters, which can be achieved when both criteria are satisfied. While strategies such as enforcing entropy can be effective in some environments, they are only heuristic strategies and should not be relied upon exclusively. Furthermore, as illustrated, both criteria may be subject to a trade-off. In more complex environments, an efficient exploration strategy may require balancing both criteria, e.g., through a schedule on the learning objective weights.

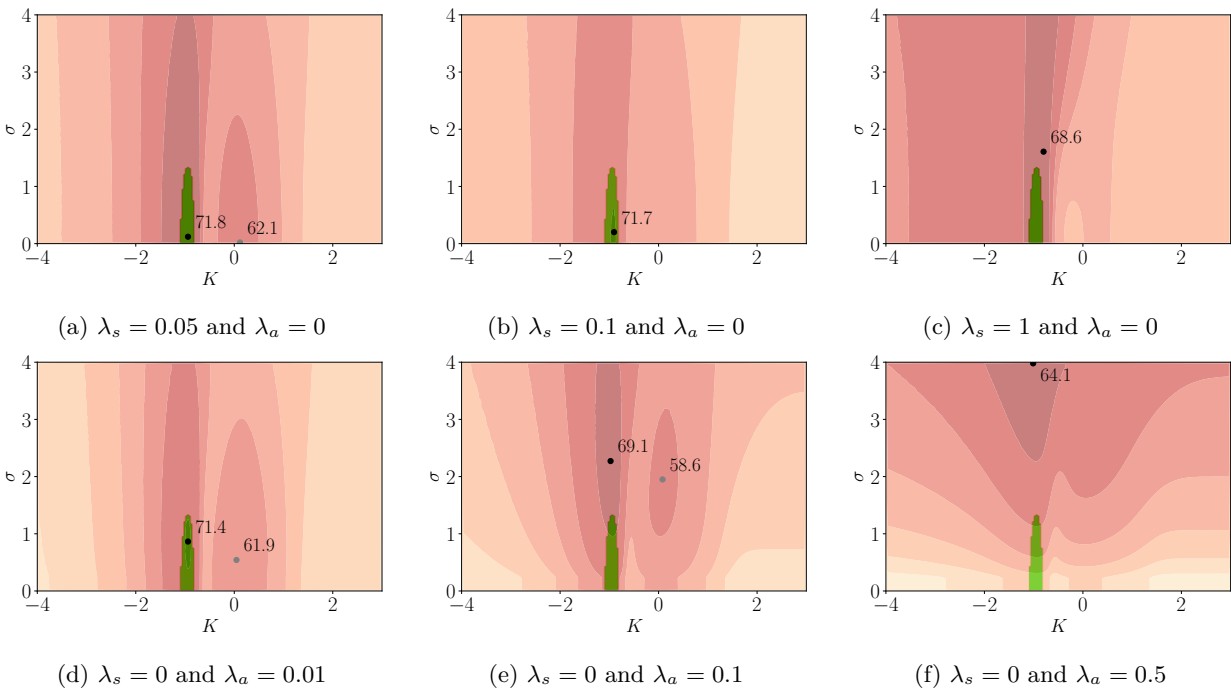

Figure 2: Contour map of (scaled) learning objective functions for different values of $\lambda_s$ and $\lambda_a$. The darker the map, the larger the learning objective value. The black dot is the parameter $\theta^\dagger$ globally maximizing the learning objective and the grey dot is the local (non-global) maximum of the learning objective if it exists. Both are labeled with the return values of the corresponding policies. The green area represents the set $\Omega = \{\theta' \,|\, \max_\theta J(\pi_\theta) - J(\pi_{\theta'}) \leq \epsilon = 1\}$.

## 4 Study of the Ascent Direction Distribution

Optimizing pseudoconcave functions with stochastic ascent methods is guaranteed to converge (at a certain rate) under assumptions about the distribution of the gradient estimates (Bottou, 2010; Chen & Luss, 2018; Ajalloeian & Stich, 2020). In this section, we study the influence of the exploration terms on this distribution in the context of policy gradients. More precisely, we study the probabilities of improving the learning objective and the return through stochastic ascent steps. Intuitively, these probabilities should be sufficiently large for the algorithm to be efficient. We formalize this intuition and illustrate how exploration can increase them, leading to more efficient algorithms.

### 4.1 Policy-Gradient Estimated Ascent Direction

In general, ascent algorithms update parameters in a direction $\hat{d}$ to locally improve an objective function $f$. The quality of these algorithms can therefore be studied (for a small step size $\alpha \to 0$) through the random variable representing the amount by which the objective increases for each $\theta$

$$X = f(\theta + \alpha\hat{d}) - f(\theta) = \alpha \langle \hat{d}, \nabla_\theta f(\theta) \rangle, \tag{7}$$

where $\langle \cdot, \cdot \rangle$ is the Euclidean scalar product. This variable depends on the random event $\hat{d}$.

The convergence of ascent algorithms is usually studied by bounding the suboptimality of $f(\theta)$ as a function of the number of ascent steps. In this active field of research, the rates are characterized by assumptions on $\hat{d}$ or, equivalently, on $X$. Depending on the assumptions, the efficiency of algorithms is, for example, studied as a function of the statistical moments of the ascent estimate or of the probability of alignment between the gradient and the estimate. We choose to work in the context in which gradient estimates have positive expected improvements $\mathbb{E}[X] > 0$ (e.g., because the estimates are unbiased), and we quantify whether this

expectation is positive due to rare realizations of large values of $X$. Intuitively, an algorithm with positive expected improvements $\mathbb{E}[X] > 0$ should theoretically converge but may be inefficient in practice if positive events $X > 0$ rarely occur. Below, we study the influence of such events, typically observed in sparse-reward environments, and the chosen context is sufficient to assess efficiency without more assumptions. Further work may proceed differently to derive convergence rates or quantify other phenomena.

In the following, we introduce two new criteria on the probability of improvement $P(X > 0)$, which we empirically validate later. First, we define an exploration strategy as $\delta$-efficient if, and only if, following the ascent direction $\hat{d} \approx \nabla_\theta L(\theta)$ has a probability of at least $\delta$ of increasing the learning objective $L(\theta)$ almost everywhere. Second, an exploration strategy is $\delta$-attractive if, and only if, there exists a neighborhood of $\theta^\dagger$ containing the parameter $\theta^{int}$ that maximizes the intrinsic return $J^{int}$, where the probability of increasing the return by following $\hat{d}$ is almost everywhere at least equal to $\delta$. Note that each probability measure and random variable are functions of $\theta$, which we do not explicitly write to keep the notation simple.

**Efficiency criterion.** *An exploration strategy is $\delta$-efficient if, and only if,*

$$\forall^\infty \theta : \mathbb{P}(D > 0) \geq \delta \,, \tag{8}$$

*where $D = \langle \hat{d}, \nabla_\theta L(\theta) \rangle$.*

**Attraction criterion.** *An exploration strategy is $\delta$-attractive if, and only if,*

$$\exists B(\theta^\dagger) : \theta^{int} \in B(\theta^\dagger) \,, \tag{9}$$

*such that*

$$\forall^\infty \theta \in B(\theta^\dagger) : \mathbb{P}(G > 0) \geq \delta \,, \tag{10}$$

*where $\theta^{int} = argmax_\theta J^{int}(\pi_\theta)$, $B(\theta^\dagger)$ is a ball centered in $\theta^\dagger$, and $G = \langle \hat{d}, \nabla_\theta J(\pi_\theta) \rangle$.*

The efficiency criterion quantifies how often a stochastic gradient ascent step improves the learning objective. The larger, the better the learning objective and its stochastic ascent direction approximations. The rationale behind the attraction criterion is that, in many exploration strategies, the intrinsic reward is dense, and it is then presumably easy to optimize the intrinsic return in the sense that $\mathbb{P}(\langle \hat{i}, \nabla_\theta J^{int}(\pi_\theta) \rangle > 0)$ is large. It implies that it is easy to locally improve the learning objective by (solely) increasing the value of the intrinsic motivation terms. It furthermore implies that policy-gradient algorithms may tend to converge towards $\theta^{int}$ rather than $\theta^\dagger$ when $\mathbb{P}(\langle \hat{d}, \nabla_\theta J(\pi_\theta) \rangle > 0)$ is small. If the attraction criterion is satisfied for large $\delta$, the latter is less likely to happen as policy gradients will eventually tend to improve the return of the policy if the parameter approaches $\theta^{int}$ and enters the ball $B(\theta^\dagger)$; eventually converging towards $\theta^\dagger$.

These two new criteria on $\hat{d}$ are independent of the previous ones on $L$, which only captured the quality of the deterministic learning objective functions. In the particular cases where the learning objectives $L$ are $\epsilon$-coherent, for $\epsilon = 0$, and pseudoconcave, e.g., with potential-based intrinsic rewards, only the distribution of estimates $\hat{d}$ can explain why some algorithms outperform others. Finally, the value of $\delta$ in the two new criteria could be related to the variance of the estimate $\hat{d}$ (e.g., via Cantelli's concentration inequalities).

## 4.2  Illustration of the Effect of Exploration on the Estimated Ascent Direction

Exploration is usually promoted and tested for problems where the reward function is sparse, typically in maze environments (Islam et al., 2019; Liu & Abbeel, 2021; Guo et al., 2021). In this section, we first introduce a new maze environment with sparse rewards to illustrate the influence of exploration on the gradient estimates of the learning objective. To this end, we present two learning objective functions and elaborate on the influence of exploration on the performance of policy-gradient algorithms in light of the efficiency and attraction criteria.

Let us consider a maze environment consisting of a horizontal corridor composed of $S \in \mathbb{N}$ tiles. The state of the environment is the index of the tile $s \in \{1, \dots, S\}$, and the actions consist of going left $a = -1$ or right $a = +1$. When an action is taken, the agent stays idle with probability $p = 0.7$, and moves with probability

$1 - p = 0.3$ in the direction indicated by the action, then $s' = \min(S, \max(1, s + a))$. The agent starts in state $s = 1$, and the target state $s = S = 15$ is absorbing. A zero reward is observed except when the agent reaches the target state, where a reward of $r = 100$ is observed. A discount factor of $\gamma = 0.99$ is considered. Finally, we study the policy that moves right with probability $\theta$ and left with probability $1 - \theta$, with density

$$\pi_\theta(a|s) = \begin{cases} \theta & \text{if } a = +1 \\ 1 - \theta & \text{if } a = -1 \,. \end{cases} \tag{11}$$

The return $J(\pi_\theta)$ is represented in black in Figure 3a as a function of $\theta$ along with two intrinsic returns, $J^a(\pi_\theta)$ in orange and $J^s(\pi_\theta)$ in blue. The intrinsic reward $\rho^a(s, a) = -\log \pi_\theta(a|s)$, from equation (3), and the intrinsic reward $\rho^s(s, a) = -\log d^{\pi_\theta, \gamma}(s)$, from equation (4) are used, respectively. In Figure 3b, we illustrate the return of the policy without exploration $J(\pi_\theta)$, along with two learning objective functions, $L^a(\theta)$ and $L^s(\theta)$, using as exploration strategies the intrinsic returns $J^a(\pi_\theta)$ and $J^s(\pi_\theta)$. We observe that the return is a pseudoconcave function with respect to $\theta$ and the optimal parameter is $\theta^* = 1$. In addition, the two learning objectives satisfy the $\epsilon$-coherence criterion for $\epsilon = 0$, implying that $\theta^* = \theta^\dagger$, and also satisfy the pseudoconcavity criterion. It is important to note that with regard to the discussion from Section 3, there is no interest in optimizing the learning objectives rather than directly optimizing the return, as the latter is already pseudoconcave. In the following, we illustrate how choosing a correct exploration strategy still deeply influences the policy-gradient algorithms when it comes to building gradient estimates.

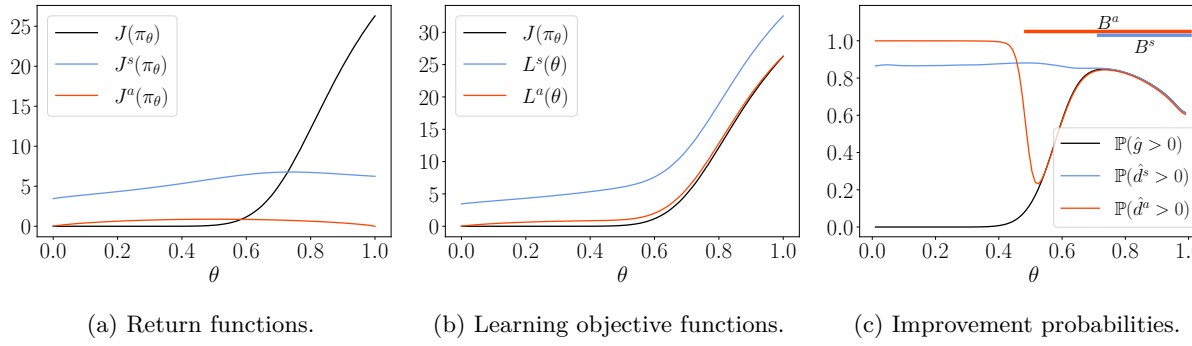

(a) Return functions.  (b) Learning objective functions.  (c) Improvement probabilities.

Figure 3: Figure 3a represents the return of the policy along with two intrinsic return functions. In Figure 3b the return is also represented together with two learning objective functions, corresponding to the two intrinsic returns. Figure 3c illustrates the probability (estimated by Monte-Carlo) of positive stochastic gradient (derivative) estimates $J(\pi_\theta)$, $L^a(\theta)$, and $L^s(\theta)$. At the top of the figure, the intervals $B^a = [\theta^{int,a}, \theta^{\dagger,a}]$ and $B^s = [\theta^{int,s}, \theta^{\dagger,s}]$ are represented. These intervals represent the smallest balls containing the parameters maximizing the intrinsic return and the learning objective, for both exploration strategies.

Let us compute the estimates $\hat{g}$ and $\hat{d}$ relying on REINFORCE (Williams, 1992) by sampling 8 histories of length $T = 100$. In this particular environment, $\mathbb{P}(D > 0)$ equals $\mathbb{P}(G > 0)$, and equals the probability that the derivative is positive. We represent in Figure 3c this probability for the return and for both learning objectives. First, we see that the learning objectives are more efficient than the return, meaning they are $\delta$-efficient for larger values of $\delta$. Depending on the parameter value, the objective $L^a(\theta)$ or $L^s(\theta)$ is best in that regard. Second, concerning the attraction criterion, we represent at the top of Figure 3c the intervals $B^a = [\theta^{int,a}, \theta^{\dagger,a}]$ and $B^s = [\theta^{int,s}, \theta^{\dagger,s}]$. They correspond to the smallest balls containing the maximizers of the intrinsic return and of the learning objective. Let the minima of the orange and blue curves over these intervals be denoted by $\delta^a$ and $\delta^s$. By definition of the attraction criterion, it is thus satisfied for any values of $\delta$ at most equal to $\delta^a$ and $\delta^s$, for $L^a(\theta)$ and $L^s(\theta)$, respectively. All these observations can eventually be explained because the computation of $\hat{g}$ is always zero when the target is not sampled in the histories, which is highly likely for policies with small values of $\theta$. Algorithms relying on intrinsic exploration would compute optimal policies efficiently where naive optimization without exploration would fail or be sample inefficient.

We have empirically shown that a well-chosen exploration strategy in policy gradients may not only remove local extrema from the objective function, but may also increase the probability that stochastic ascent steps

improve the objective function and eventually the return. Under the previous assumptions, this probability measures the efficiency of algorithms. Furthermore, among different learning objectives respecting the coherence and pseudoconcavity criteria, it is best to choose one that has high values for $\delta$ in both the efficiency and attraction criteria. In Section 5, we extend the study to more complex environments and policies.

The problem discussed in this section strongly relates to overfitting or generalization in reinforcement learning. In situations where the same state and action pairs are repeatedly sampled with high probability, the policy may appear optimal by neglecting the rewards observed in state and action pairs sampled with low probability. The gradient estimates will then be zero with high probability, and the gradient updates will not lead to policy improvements. In the previous example, gradient estimates computed from policies with a small parameter value $\theta$ wrongly indicate that a stationary point has been reached as they equal zero with high probability. We quantify this effect with a novel definition of local optimality. We define as locally optimal policies over a space with probability $\Delta$ the policies that maximize the reward on expectation over a set of states and actions observed in a history with probability at least $\Delta$. Formally, a policy $\pi$ is locally optimal over a space with probability $\Delta$ if, and only if,

$$\exists \mathcal{E} \in \left\{ \mathcal{X} \Big|_{\substack{s \sim d^{\pi,\gamma}(\cdot) \\ a \sim \pi(\cdot|s)}} \mathbb{E} \left[ \mathbb{1}\left[ (s,a) \in \mathcal{X} \right] \right] \geq \Delta \right\} : \pi \in \arg\max_{\pi'} \mathbb{E}_{\substack{s \sim d^{\pi',\gamma}(\cdot) \\ a \sim \pi'(\cdot|s)}} \left[ \mathbb{1}\left[ (s,a) \in \mathcal{E} \right] \rho(s,a) \right] . \tag{12}$$

In the case of sparse rewards, many policies observe with high probability state and action pairs with zero rewards and are locally optimal for large probabilities $\Delta$. Typically, in the previous example, the joint set $\{1, \ldots, S-2\} \times \{-1, 1\}$ is a set of state and action pairs $\mathcal{E}$ that satisfies the definition in equation (12) for large values $\Delta$ when $\theta$ is small. As we have shown, exploration mitigates the convergence of policy-gradient algorithms towards these locally optimal policies. Note that assuming a non-zero reward is uniformly distributed over the state and action space, exploration policies with uniform probabilities over visited states and actions are the best choice for sampling non-zero rewards with high probability. It can thus be considered the best choice of exploration to reduce the probability that the stochastic gradient ascent steps do not increase the objective value. Such initial policy may be learned as suggested by Lee et al. (2019).

## 5 Extended Experiments

In this section, we introduce complex environments and parameterize policies with neural networks. We then learn the policy parameters using policy gradients and intrinsic exploration bonuses. We discuss convergence results based on the four criteria introduced in the paper. In this context, it is impractical to naively evaluate the different criteria for each policy parameter, and we are restricted to a local evaluation of the criteria on the basis of statistics computed during optimization.

Let us adapt environments from the MiniGrid suite of environments (Chevalier-Boisvert et al., 2023). Each environment corresponds to a maze in which an agent moves, aiming to reach a target position. The agent may choose as actions to turn left, turn right, move forward, or stay idle. We consider two reward settings: dense and sparse. In the first, rewards of $-1$ are received for every non-idle move, and a reward of $1000$ is received upon reaching the target position. In the second setting, zero rewards are received everywhere, except upon reaching the target position, where a bonus of $1000$ is provided. We consider $\gamma = 0.98$.

In the following, the policy is parameterized with a fully connected neural network taking as input the position pair and the orientation of the agent, and outputting a categorical distribution over actions. The network is composed of three hidden layers of 64 neurons with ReLU activation functions. The policy is optimized in the different environments and reward settings. To that end, we consider three learning-objective functions: $J(\pi_\theta)$, $L^a(\theta)$, and $L^s(\theta)$, respectively with $\lambda_a = 0.5$ and $\lambda_s = 0.25$. For the last objective, the state-visitation density estimator is a ten-component Gaussian mixture model that maximizes the likelihood of the sampled batch. The optimization is performed using the Adam update rule (Kingma & Ba, 2014), with REINFORCE ascent directions computed over 32 histories of constant length $T = 100$, and with a learning rate (step size) equal to $0.0005$. The length $T$ of the histories is chosen such that the realized value of $T$ from a geometric distribution with success probability parameter $1 - \gamma$ has at least a cumulative probability of $0.85$.

We first consider the environments in the dense-reward setting and discuss convergence in light of the pseudoconcavity and $\epsilon$-coherence criteria. In Figure 4, we provide the evolution of the return of the policies when optimizing the three objectives $J(\pi_\theta)$, $L^a(\theta)$, and $L^s(\theta)$ for the different environments. On the one hand, in the `MiniGrid-Empty-8x8-v0` and `MiniGrid-FourRooms-v0` environments, optimizing the return provides high-performing policies, and the policies resulting from the optimization of the other learning objectives are suboptimal. The difference in performance is justified by the $\epsilon$-coherence criterion, where $\epsilon$ is the bound on the best policy that can be found when optimizing the learning objective (assuming that the global optimum of each objective is found). On the other hand, in the other environments, optimizing the return often leads to convergence to locally optimal parameters (or saddle points). These parameters correspond to idling policies and are locally optimal due to the action penalization incurred when moving. The policies obtained when optimizing $L^a(\theta)$ and $L^s(\theta)$ no longer fall into the previous local optima. The latter illustrates the validity of the pseudoconcavity criterion in that region of the parameter space. Note that when optimizing $L^a(\theta)$, the policy tends to converge towards another local optimum where actions are uniformly distributed. We nevertheless hypothesize that it is a local optimum in the sense of equation (12).

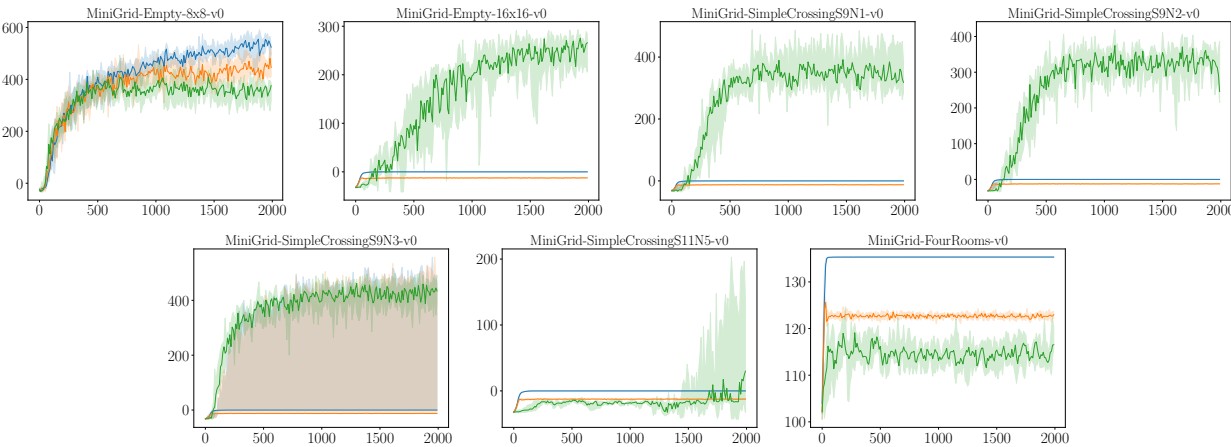

Figure 4: Return of policies during optimization in the dense minigrid environments. In blue, the return $J(\pi_\theta)$ is optimized; in orange, the learning objective $L^a(\theta)$ is optimized; and in green, the learning objective $L^s(\theta)$ is optimized by performing Adam steps in REINFORCE directions. Note that the median, worst, and best cases over five runs are represented for the different curves.

In the previous experiments, the local optima arise due to the negative rewards associated with idle actions. If we consider the sparse-reward setting, we could then assume that directly optimizing the return is sufficient to find high-performing policies. We use the same parameters as previously and provide the evolution of the return of the policies when optimizing the three objectives $J(\pi_\theta)$, $L^a(\theta)$, and $L^s(\theta)$ for the different environments in Figure 5. On the one hand, for most environments, regardless of the learning objective, the resulting policy has a high return. Note that the $\epsilon$-coherence can again be illustrated, as the policies resulting from the optimization of the return perform better than the others. On the other hand, for the `MiniGrid-Empty-16x16-v0` and `MiniGrid-SimpleCrossingS11N5-v0` environments, a high-performing policy can only be found when optimizing the learning objective $L^s(\theta)$. We illustrate in the following that these results can be justified by the efficiency and attraction criteria.

In Figure 6a, we show the probability of improving the return $J(\pi_\theta)$ and the learning objective $L^s(\theta)$ for each parameter obtained during their respective optimization by stochastic gradient ascent steps (i.e., SGA following respectively $\hat{g} \approx \nabla_\theta J(\pi_\theta)$ and $\hat{d} \approx \nabla_\theta L^s(\theta)$). The improvement probability is higher for the learning objective than for the return. In other words, for these parameters, the efficiency $\delta$ of the learning objective is much higher than that of the return, which justifies the failure of optimization without exploration in Figure 5. In Figure 6b, we again show the probability of improving the return $J(\pi_\theta)$ and the learning objective $L^s(\theta)$, but for each parameter obtained during the stochastic gradient ascent steps on $L^s(\theta)$ (i.e., SGA following $\hat{d} \approx \nabla_\theta L^s(\theta)$). The probability of improving the return when optimizing the

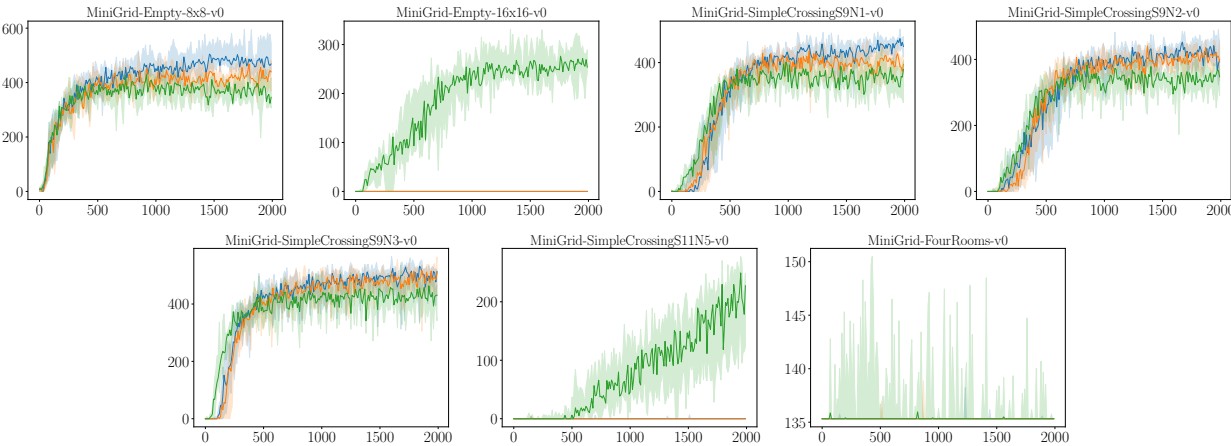

Figure 5: Return of policies during optimization in the sparse minigrid environments. In blue, the return $J(\pi_\theta)$ is optimized; in orange, the learning objective $L^a(\theta)$ is optimized; and in green, the learning objective $L^s(\theta)$ is optimized performing Adam steps in REINFORCE directions. Note that the median, worst, and best cases over five runs are represented for the different curves.

learning objective is small at the beginning and increases after some iterations. This indicates that after some iterations, the attraction criterion holds for a large $\delta$. The return thus eventually starts increasing, and optimizing $L^s(\theta)$ provides a high-performing policy, as can be seen in Figure 5.

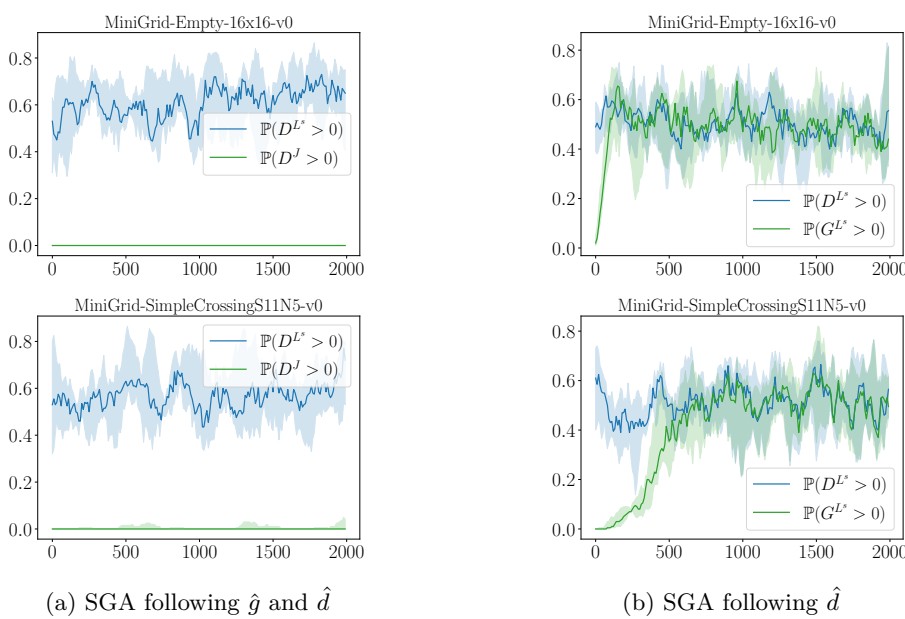

(a) SGA following $\hat{g}$ and $\hat{d}$        (b) SGA following $\hat{d}$

Figure 6: In Figure 6a, we show the estimated probability of improving the return $J(\pi_\theta)$ and the learning objective $L^s(\theta)$ when following their REINFORCE gradient estimates. In Figure 6b, we show the estimated probability of improving the return $J(\pi_\theta)$ and the learning objective $L^s(\theta)$ when following the REINFORCE gradient estimate of the learning objective $L^s(\theta)$. Probabilities were estimated based on the frequencies of improving the objective functions by more than 0.2 after following 5 Adam ascent steps.

# 6 Conclusion

In conclusion, this research takes a step toward dispelling misunderstandings about exploration through the study of its effects on the performance of policy-gradient algorithms. More specifically, we distinguish two effects exploration has on the optimization. First, it modifies the learning objective to remove local extrema. Second, it modifies the gradient estimates and increases the likelihood that the update steps lead to improved returns. These two phenomena are studied through four criteria that we introduce and illustrate.

These ideas apply to other direct policy optimization algorithms. Indeed, the four criteria do not assume any structure on the learning objective and can thus be straightforwardly applied to any objective function optimized by a direct policy search algorithm. In particular, for off-policy policy-gradient methods, we may simply consider that the off-policy objective is itself a surrogate or that the gradients of the return are biased estimates based on past histories. Ideas introduced in this work also apply to other reinforcement learning techniques. Typically, for value-based RL with sparse-reward environments, convergence to a value function that outputs zero is expected with high probability. This is mostly due to the low probability of sampling non-zero rewards via Monte Carlo. Similar discussion and analysis as in Section 4 then apply.

Our framework opens the door to further theoretical analysis, and the potential development of new criteria. We believe that deriving practical conditions on the exploration strategies and the scheduling of the intrinsic return to guarantee fast convergence should be the focus of attention. It could be achieved by bounding the policy improvement in expectation, which is nevertheless usually a hard task without strong assumptions. We further believe that we provide a new lens on exploration that is necessary for interpreting and developing exploration strategies, in the sense of optimizing surrogate learning objective functions.

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
