# OpenReview forum: "Behind the Myth of Exploration in Policy Gradients"
_TMLR — Rejected by TMLR_

### Review · Reviewer_Suoe · 2025-07-09

**Summary Of Contributions:**

This paper studies how exploration bonuses work in policy gradient reinforcement learning. Instead of just viewing them as a way to help agents explore the environment better, the authors analyze their role from an optimization perspective. They introduce four criteria—two related to the learning objective (coherence and pseudoconcavity), and two related to the stochastic ascent direction (efficiency and attraction). These criteria aim to explain how intrinsic reward shaping influences convergence in policy-gradient algorithms. They also provide experiments and examples to support their analysis.

**Audience:**

Yes

**Broader Impact Concerns:**

No ethical or broader impact concerns.

**Claims And Evidence:**

Yes

**Requested Changes:**

See above.

**Strengths And Weaknesses:**

# Strengths:
1. The four criteria (coherence, pseudoconcavity, efficiency, and attraction) offer a clear and structured way to understand why exploration bonuses often help in training. It’s a valuable shift from relying only on intuition to having formal tools for analysis.

2. The trade-off between making the optimization smoother (pseudoconcavity) and staying close to the original goal (coherence) is explained well.

3. They nicely analize the idea in both simple and complex environments like maze, hill and MiniGrid.

4. The paper is well-organized and the visuals are helpful. The figures clearly show how exploration affects the learning process.

# Weaknesses:
1. The paper focuses mainly on entropy-based and visitation-based rewards. It would be great to see how the framework applies to other popular methods like curiosity or RND. Even a short discussion would help.

2. The experiments use fixed values for the exploration weights (λs, λa), but don’t really explore how sensitive the results are to these choices. Some kind of ablation study could make the conclusions stronger.

3. It would be helpful to understand how well this approach generalizes beyond grid-based navigation tasks. For instance:

- Can the intrinsic objectives (e.g., La(θ), Ls(θ)) and density estimators be used as-is in continuous action spaces or non-episodic settings?
- Are there environment-specific assumptions—like episodic structure, state discretization, or known goal states—needed to compute the intrinsic bonuses?
- What modifications, if any, would be needed to apply this framework to more complex domains like MuJoCo, Atari, or robotics?

4. A bit more discussion about how this framework might extend to off-policy or actor-critic settings would be useful.

5. The paper does not explicitly address whether the same exploration principles can be applied in dynamic or continuous-control environments. A short discussion on it would be useful.

---

> ### Author Response · Authors · 2025-09-09
> **Response to review**
>
> Thank you for your review. We agree that several remarks point to limits of our current analysis, and we will emphasize them. Below is a point by point response with proposed updates. We believe only minor edits are needed and we kindly ask for your support in accepting the paper.
>
> > The paper focuses mainly on entropy-based and visitation-based rewards. It would be great to see how the framework applies to other popular methods like curiosity or RND. Even a short discussion would help.
>
> Our focus on entropy and visitation bonuses was deliberate because they do not depend on the learning dynamics and are easier to illustrate. Uncertainty based methods tie the intrinsic reward to what the agent has learned so far, which makes clean analysis harder. The ideas are closely related. Uncertainty is typically high in rarely visited regions, where visitation entropy is low. Our criteria apply in the same way, but the uncertainty signal evolves as data accumulates. We will clarify this relationship and explain our choice in the background section.
>
> > The experiments use fixed values for the exploration weights (λs, λa), but don’t really explore how sensitive the results are to these choices. Some kind of ablation study could make the conclusions stronger
>
> We agree that the choice and schedule of the exploration weights influence the results. The weights control the tradeoff between pseudoconcavity and coherence, which we illustrate in the first experiments. In sparse reward settings a small weight often suffices. Early in training the extrinsic return is flat, so the intrinsic signal drives progress even with a small weight. Once nonzero returns appear, a small weight preserves coherence rather than over shaping the landscape. Because these effects depend on the task and estimator, a general sensitivity law is hard to state. We will emphasize this discussion and note scheduling weights are valuable directions for future work.
>
> > It would be helpful to understand how well this approach generalizes beyond grid-based navigation tasks. For instance: 1) Can the intrinsic objectives and density estimators be used as-is in continuous action spaces or non-episodic settings? 2) Are there environment-specific assumptions—like episodic structure, state discretization, or known goal states—needed to compute the intrinsic bonuses? 3) What modifications, if any, would be needed to apply this framework to more complex domains like MuJoCo, Atari, or robotics?
>
> Sparse-reward tasks are often grid based in the literature and already make exploration hard on simple benchmarks. We expect the same phenomenon in continuous settings with sparse rewards, but standard benchmarks are less common to our knowledge.
>
> In continuing RL the objective changes. Returns are usually measured under the stationary distribution, not the discounted visitation measure, so initial states do not matter. The framework is general, and similar conclusions apply, but exploration methods should be adapted to this objective.
>
> Overall, the criteria and discussion remain valid, but they are not yet practical to verify efficiently. Future work will provide more usable sufficient conditions that can be checked beforehand. We will emphasize these limits and directions in the paper.
>
> > A bit more discussion about how this framework might extend to off-policy or actor-critic settings would be useful.
>
> We agree this extension is useful. Our framework does not assume a specific surrogate, so it can be adapted. In off-policy settings one can view the off-policy gradient as a biased estimate of the on-policy gradient, or adopt a different objective and repeat the discussion. The same split holds between effects on the learning objective and on the gradient estimate. For actor-critic, if the critic has no learning error, the same arguments apply. Learning the dynamics can add extra effects such as randomness that may aid exploration.
>
> The main limitation is that it is hard to illustrate the criteria exactly in these settings. Future work should develop more practical criteria and study off-policy, continuing RL, and actor-critic in focused test cases. We will emphasize these limits and the need for follow-up work in the paper.
>
> > The paper does not explicitly address whether the same exploration principles can be applied in dynamic or continuous-control environments. A short discussion on it would be useful.
>
> We focus on classical stationary MDPs. Extending the analysis to dynamic non stationary or continuous time settings is outside our scope. We expect similar phenomena, but a careful treatment is needed. We will add a short note and list this as future work.

---

> > ### Comment · Reviewer_Suoe · 2025-09-11
> >
> > ## 1. Ablation Study
> > While the explanation about the choice and scheduling of exploration weights clarifies their role in balancing pseudoconcavity and coherence, an **ablation study or sensitivity analysis** would strengthen the conclusions. Showing how varying these weights affects learning performance across tasks would provide stronger empirical support for the theoretical claims.
> >
> > ## 2. Sparse-Reward Continuous-Control Tasks
> > Regarding the statement:
> >
> > > “Sparse-reward tasks are often grid based… standard benchmarks are less common to our knowledge.”
> >
> > It is important to note that **sparse-reward tasks in continuous-control benchmarks are well established** in recent literature. Some examples include:
> >
> > - **FuRL (2024):** *Visual-Language Models as Fuzzy Rewards for Reinforcement Learning*, applied on Meta-World sparse-reward manipulation tasks using continuous actions (Fu et al., 2024).
> > - **Deep Exploration with PAC-Bayes (Tasdighi et al., 2025):** uses MuJoCo continuous-control tasks with sparse rewards, as well as Metaworld sparse reward tasks in actor-critic settings.
> > - **Other examples in D4RL / Sparse MuJoCo:** e.g., SparseHalfCheetah, AntMaze, which are standard sparse-reward continuous-control benchmarks.
> >
> > These references suggest that sparse-reward continuous-action benchmarks are **not as rare or grid-based as implied**.
> >
> > Therefore, I remain **not fully convinced** by some of the authors’ justifications:
> >
> > 1. An **ablation or sensitivity analysis** on exploration weight choices is still missing.
> > 2. The discussion around sparse-reward continuous-control benchmarks seems **incomplete or somewhat inaccurate**, as multiple well-established examples exist.
> >
> > Providing more empirical support and acknowledging existing benchmarks would make the paper more convincing.
> >
> > ## References
> > - Fu, Y., Zhang, H., Wu, D., Xu, W., Boulet, B. (2024). *FuRL: Visual-Language Models as Fuzzy Rewards for Reinforcement Learning*. Proceedings of PMLR, 235.
> > - Tasdighi, B., Haussmann, M., Werge, N., Wu, Y.-S., Kandemir, M. (2025). *Deep Exploration with PAC-Bayes*. ECAI-2025

---

> > > ### Author Response · Authors · 2025-09-11
> > > **Clarification sensitivity analysis and continuous control**
> > >
> > > Thank you for your time. Please find below a clarification of our previous response.
> > >
> > > > An ablation or sensitivity analysis on exploration weight choices is still missing.
> > >
> > > We already report a sensitivity analysis for the first two criteria in Fig. 2. Extending this to the remaining two criteria is straightforward: it amounts to forming the relevant linear combinations of the returns reported in Fig. 3(a) and providing the corresponding event probabilities in Fig. 3(c) for different weights. We can add these plots and summarize our current discussion in the paper. Our preliminary runs match the discussion: a sufficiently small weight improves convergence, while a large weight harms coherence and can create or accentuate local maxima.
> > >
> > > > The discussion around sparse-reward continuous-control benchmarks seems incomplete or somewhat inaccurate, as multiple well-established examples exist.
> > >
> > > You are right that our earlier response was incomplete. We will acknowledge and cite established sparse-reward continuous-control benchmarks. The benchmarks you highlighted likely require complex policy parameterizations with locally optimal behaviors, which makes it hard to isolate the last two criteria from the first two. We will clarify this limitation. A fuller evaluation in these MDPs would surely be valuable but is also challenging. For the above-mentioned reasons, we believe additional experiments would not be more conclusive and would lead to similar analyses as in our supplementary experiment section.

---

### Review · Reviewer_TrY2 · 2025-07-28

**Summary Of Contributions:**

The authors explain how reward shaping (adding intrinsic rewards to the value function) may aid in the exploration for PG algorithms. They specifically identify the conditions that lead to convergence to the global optimum policy parameters without getting stuck in the stationary points.

**Audience:**

No

**Claims And Evidence:**

Yes

**Requested Changes:**

1. Some existing literatures, such as [1], establish an optimal convergence rate for the PG algorithms without reward shaping. Can the authors (theoretically) demonstrate how the surrogate objective provides better convergence in comparison to the above algorithms?

2. Is it possible to identify the class of intrinsic rewards that the agents can adaptively choose based on some observations? Note that ensuring that the surrogate objective remains pseudo-concave is hard to ensure without knowing some global properties of the value functions. This is impossible in most RL scenarios since the agent is unaware of the state-transition kernel, and therefore cannot compute the exact values of the objective functions for different policy parameters, let alone infer its global properties.

[1] Mondal, W. U., and V. Aggarwal. "Improved sample complexity analysis of natural policy gradient algorithm with general parameterization for infinite horizon discounted reward Markov decision processes." International Conference on Artificial Intelligence and Statistics. PMLR, 2024.

**Strengths And Weaknesses:**

**Strength**: The authors identify four mathematical conditions (two for the objective function and two for the stochastic gradient directions) that ensure the convergence of the PG algorithms to the global maximum. The analysis is generic, i.e., not tailored for any specific PG algorithm.

**Weakness**: Although the study might provide some insights, its relevance to theoretical analysis or practical applicability remains unclear.

1. The authors mention that if the surrogate objective function is pseudo-concave (i.e., does not have any stationary point other than the global maximum) and $\epsilon$-coherent (i.e., the value function computed at its maximum point is within $\epsilon$ distance of the optimal value), then the PG-algorithm is likely to converge to the neighbourhood of its desired optimum. This is quite obvious. However, the implication of this insight is not evident. How can one choose the intrinsic value functions such that the surrogate objective satisfies the stated criteria? Surely, this is impossible to do in general because, for many cases (e.g., neural network-based policies with large depth/width), the dimension of the parameter space is too large to comment on any specific global property of the value function.

2. Similarly, it is impossible to comment on any specific properties of the stochastic gradient of the surrogate objective function in general. Therefore, how do the insights provided by the authors on the desired properties of the gradients help design better algorithms?

In light of the above comments, it seems that the study does not add any value to the RL literature. In addition to the above comments, I would like to mention the following minor comments.

3. The definition of pseudo-concavity as given in (6) seems to be problematic. The authors intend to define a function to be pseudo-concave if it has a unique maximum and no other stationary point. However, this is not captured by (6). In particular, (6) says that a function is pseudo-concave if its global optimum point is not stationary.

4. The potential-based reward function (used in the consistency theorem) is not explicitly defined in the main text.

---

> ### Author Response · Authors · 2025-09-09
> **Response to review (part 1)**
>
> Thank you for your review. Below is a point by point response on the few limitations you highlighted. We believe only minor edits are needed. We are confident the paper meets TMLR acceptance criteria and we kindly ask for your support in accepting the paper.
>
> > The authors mention that if the surrogate objective function is pseudo-concave (i.e., does not have any stationary point other than the global maximum) and \eps-coherent (i.e., the value function computed at its maximum point is within  distance of the optimal value), then the PG-algorithm is likely to converge to the neighbourhood of its desired optimum. This is quite obvious. However, the implication of this insight is not evident. How can one choose the intrinsic value functions such that the surrogate objective satisfies the stated criteria? Surely, this is impossible to do in general because, for many cases (e.g., neural network-based policies with large depth/width), the dimension of the parameter space is too large to comment on any specific global property of the value function.
> > Similarly, it is impossible to comment on any specific properties of the stochastic gradient of the surrogate objective function in general. Therefore, how do the insights provided by the authors on the desired properties of the gradients help design better algorithms?
>
> We agree that convergence with true gradients under the two first criteria is intuitive. Our aim was to state minimal conditions needed to guaranteed in general. We then show that these conditions hold for some intrinsic rewards. We believe we are first to clarify this effect. Verifying the criteria a priori can be harder than solving the optimization itself, so they are not meant as a practical test.
>
> The framework is intended to clarify the effect of exploration from an optimization viewpoint. It separates two roles. One is shaping the objective to avoid spurious optima (the one you discuss in your review). The other is improving sample efficiency for stochastic ascent (one you did not highlight in your review). This separation helps interpretation and guides empirical choices, even without full guarantees. We believe, we are first to do the distinction, as it is obvious for the reviewer, we kindly ask to point out to references. Furthermore, highlighting limitations of the litterature and distinguishing cofounding effects is considered valid contributions to TMLR from our understanding.
>
> For entropy regularization, global convergence results exist under PL-type assumptions, which are closely related to the pseudoconcavity condition we discuss. Identifying sufficient and checkable conditions that practitioners can use is an open direction that we highlight as future work. In high-dimensional settings it is unrealistic to certify global properties, but the framework helps focus analysis and experiment design on the two effects above.
>
> > In light of the above comments, it seems that the study does not add any value to the RL literature.
>
> As the above comments only hold for the two first criteria out of four, they can not in any circumstance be considered valid to argue that the paper is of no value to the RL litterature. Furthermore, we believe the previous remark clarify some of your concerns.
>
> > The definition of pseudo-concavity as given in (6) seems to be problematic. The authors intend to define a function to be pseudo-concave if it has a unique maximum and no other stationary point. However, this is not captured by (6). In particular, (6) says that a function is pseudo-concave if its global optimum point is not stationary.
>
> Equation (6) defines an objective L as pseudoconcave, if and only if, there exists a unique $\theta^\dagger$ that is a stationary point (zero gradient) and that corresponds to the maximum of the objective function. It does not claim that the maximizer is not stationary. We can clarify it by avoiding the use of the logic $\land$ and using plain words.
>
> There is nevertheless a simplification compared to the real definition of pseudoconcavity, and we propose to clarify it in the footnote:
>
> "The literature on policy-gradient algorithms usually relies on the Polyak–Łojasiewicz inequality and gradient dominance (Polyak, 1963; Lojasiewicz, 1963; Karimi et al., 2016) rather than pseudoconvexity (Mangasarian, 1975), all of which imply that every stationary point is a global minimum (invexity), from which global convergence of gradient descent methods follows (Ben-Israel and Mond, 1986; Mishra and Giorgi, 2008; Bottou, 2010). Gradient dominance furthermore ensures linear rates of convergence (Karimi et al., 2016). Under quasiconvexity, invexity and pseudoconvexity coincide (Mishra and Giorgi, 2008) and are thus implied by gradient dominance. For the sake of keeping the discussion simple, the definitions of pseudoconvexity and pseudoconcavity are simplified, and additional assumptions on the stochastic gradient estimates are neglected."

---

> ### Author Response · Authors · 2025-09-09
> **Response to review (part 2)**
>
> > The potential-based reward function (used in the consistency theorem) is not explicitly defined in the main text.
>
> We will add an explicit definition and restate the consistency theorem in the main text. A reward is potential based if there exists a function $\Phi$ such that $\rho(s, a) = - \Phi(s) + \gamma \mathbb{E}_{s' \sim p(\cdot|s, a)} \Phi(s')$.
>
> > Some existing literatures, such as [1], establish an optimal convergence rate for the PG algorithms without reward shaping. Can the authors (theoretically) demonstrate how the surrogate objective provides better convergence in comparison to the above algorithms?
>
> Prior work gives optimal rates for policy gradient under conditions like gradient dominance or the PL inequality on the original objective. Our analysis shows that the same type of guarantees hold when these conditions (here, pseudoconcavity) hold for the surrogate objective. This can be easier to satisfy in some settings because the surrogate can be designed through reward shaping. We will clarify this point in the discussion and cite the relevant papers. We do not claim strictly better rates. Our contribution is that shaping can make the required conditions easier to meet.
>
> Overall it is unclear if the reviewer wants to highlight that reward shaping is of no use due to [1]. We beleive it has not been much studdied and discussed.
>
> > Is it possible to identify the class of intrinsic rewards that the agents can adaptively choose based on some observations? Note that ensuring that the surrogate objective remains pseudo-concave is hard to ensure without knowing some global properties of the value functions. This is impossible in most RL scenarios since the agent is unaware of the state-transition kernel, and therefore cannot compute the exact values of the objective functions for different policy parameters, let alone infer its global properties.
>
> The scope of this paper was to identify the first general criteria under which convergence can be achieved. Finding conditions that can be easily checked and identifying an adaptive class of intrinsic rewards is valuable but falls into further works.

---

### Review · Reviewer_njxY · 2025-08-27

**Summary Of Contributions:**

The paper empirically studies the effect of exploration bonuses on the optimization landscape of policy optimization. The authors consider two entropy-based bonuses based on the state-visitation measure and the policy. Different criteria are proposed to assess how different the regularized objective is compared to the original discounted reward. Some of these criteria are easy to estimate, but others are quite challenging. Numerical illustrations nicely demonstrate different effects the regularization may have on the optimization landscape and the training.

**Audience:**

Yes

**Broader Impact Concerns:**

While I find some illustrations and intuitions in this paper useful, I find that the methodology used is questionable, and there is no convincing evidence. After reading the paper, I am not convinced how the "framework" proposed in this work helps the future theoretical analysis of exploration in policy gradient methods.

**Claims And Evidence:**

No

**Requested Changes:**

There is no legend on Figures 4 and 5.

**Strengths And Weaknesses:**

Weaknesses:

1) The methodology is too generic and does not use the structure of discounted cumulative reward and policy parameterization. In particular, all criteria defined in this paper (5), (6), (8), (9), can be defined for a general optimization problem with regularization.

2) A better term instead of 'pseudo concavity' might be 'invexity', which is a well-established and more precise notion. However, if there is a difference between these notions, it is worth explaining.

3) One message coming from experiments on page 8 is that the exploration bonus helps optimization. However, it is unclear if this message can be generalized or if it is just a coincidence. Can authors show for any regularization parameter that there is a guarantee that optimization is faster/easier/removes local minima, or is this merely an empirical observation without any guarantees? I ask about "any" regularization parameter since for large enough, the latter question regarding local minima seems obvious that the problem will be invex even if the original problem had spurious minima. However, guarantees for "some" $\lambda_i$ may also be interesting.

4) On page 10, the authors say that 'the difference in performance is justified by \eps-coherence criterion'; however, I do not understand how this is justified. Is there any guarantee that \eps is large for these environments? Is there a convincing empirical evidence? The methodology of how to measure \esp is unclear to me.

Questions:

1) In abstract, what does "to smooth the learning objective" mean? It is already smooth. Were the authors meant to say convexity?

2) I have some doubts about the Consistency Theorem. Imagine that the objective and regularization are well aligned, i.e., the same functions. In that case, the regularization doesn't affect the optimization landscape (it only does by the scaling). The coherence condition will be satisfied with \eps = 0, but obviously $J(\pi_{\theta}) \neq L(\theta).$

---

> ### Author Response · Authors · 2025-09-09
> **Response to review (part 1)**
>
> Thank you for your review. Below is a point by point response that clarifies our contributions. We believe only minor edits are needed. We are confident the paper meets TMLR’s acceptance criteria and we kindly ask for your support in accepting the paper.
>
> > The methodology is too generic and does not use the structure of discounted cumulative reward and policy parameterization. In particular, all criteria defined in this paper (5), (6), (8), (9), can be defined for a general optimization problem with regularization.
>
> Yes, the methodology is generic and this is intentional. We state in the paper that our goal is to study exploration bonuses as surrogate objectives through the lens of numerical optimization. By abstracting away RL specifics, we can propose criteria that apply broadly and that also cover the policy gradient setting we study. This lets us make general statements about when exploration bonuses may help without committing to a particular parameterization or reward structure. For a focused study of a specific bonus and a specific algorithm additional assumptions would be needed, and we agree such specializations are valuable, yet out of the scope of this paper. At this stage it is unclar why a generic framework is considered a weakness?
>
> > A better term instead of 'pseudo concavity' might be 'invexity', which is a well-established and more precise notion. However, if there is a difference between these notions, it is worth explaining.
>
> Thank you for the suggestion. We used simplified definitions because gradient methods can converge under several standard assumptions, and pseudoconcavity is more familiar than invexity. We will clarify the relation in a footnote as follows:
>
> "The literature on policy-gradient algorithms usually relies on the Polyak–Łojasiewicz inequality and gradient dominance (Polyak, 1963; Lojasiewicz, 1963; Karimi et al., 2016) rather than pseudoconvexity (Mangasarian, 1975), all of which imply that every stationary point is a global minimum (invexity), from which global convergence of gradient descent methods follows (Ben-Israel and Mond, 1986; Mishra and Giorgi, 2008; Bottou, 2010). Gradient dominance furthermore ensures linear rates of convergence (Karimi et al., 2016). Under quasiconvexity, invexity and pseudoconvexity coincide (Mishra and Giorgi, 2008) and are thus implied by gradient dominance. For the sake of keeping the discussion simple, the definitions of pseudoconvexity and pseudoconcavity are simplified, and additional assumptions on the stochastic gradient estimates are neglected."
>
> > One message coming from experiments on page 8 is that the exploration bonus helps optimization. However, it is unclear if this message can be generalized or if it is just a coincidence. Can authors show for any regularization parameter that there is a guarantee that optimization is faster/easier/removes local minima, or is this merely an empirical observation without any guarantees? I ask about "any" regularization parameter since for large enough, the latter question regarding local minima seems obvious that the problem will be invex even if the original problem had spurious minima. However, guarantees for "some" may also be interesting.
>
> Our experiments indicate that exploration bonuses can make optimization easier, but we do not claim guarantees for every regularization weight. We focus on two minimal conditions for gradient ascent to succeed. The surrogate and the return should share the same global maximizer. The landscape should not contain spurious local optima. We verify these conditions in our settings for some values of the regularization, which implies that gradient ascent can reach the global optimum. For stochastic ascent the outcome also depends on the distribution of the estimate. It can be measured through variance. We nevertheless study a general criterion that limits failures due to rare samples and we illustrate it. In the extend experiments we report that this criterion holds in more complex environments. Other choices are possible if one wants to target a specific effect.
>
> In general one cannot ensure convexity for finite regularization weights. Adding a convex term to a nonconvex function does not make it convex in general. For example, let $f(x) = - x^4$ and $g(x) = x^2$. Then $h(x) = f(x) + \lambda g(x)$ is not convex for any finite  $\lambda$. The practical goal and the minimal requirement from theory is to avoid spurious local optima and to keep the surrogate aligned with the return. Our results show that exploration can help meet these conditions for some ranges of regularization.

---

> ### Author Response · Authors · 2025-09-09
> **Response to review (part 2)**
>
> > On page 10, the authors say that 'the difference in performance is justified by \eps-coherence criterion'; however, I do not understand how this is justified. Is there any guarantee that \eps is large for these environments? Is there a convincing empirical evidence? The methodology of how to measure \esp is unclear to me.
>
> By definition, $\epsilon$ is the minimum expected return gap when the surrogate is optimized to its global maximum. In our experiments we assume each learning curve reaches the global optimum of its objective. Any remaining performance gap in return is therefore attributed to $\epsilon$.
>
> > In abstract, what does "to smooth the learning objective" mean? It is already smooth. Were the authors meant to say convexity?
>
> By “smooth” we meant reducing the number of spurious local optima and making the landscape "more convex". The term is indeed misleading and we will replace it with clearer wording and avoid this ambiguity in the paper.
>
> > I have some doubts about the Consistency Theorem. Imagine that the objective and regularization are well aligned, i.e., the same functions. In that case, the regularization doesn't affect the optimization landscape (it only does by the scaling). The coherence condition will be satisfied with \eps = 0, but obviously $J \neq L$.
>
> Your example is a special case where the regularizer equals the objective, so scaling does not change the landscape. In that single MDP the coherence condition holds with $\epsilon = 0$ even though J and L differ by scale. Our theorem concerns uniform guarantees across environments. Prior work we cite shows there is no regularizer with $J \neq L$ that guarantees $\epsilon = 0$ for every MDP.

---

### Decision · Action_Editor_4oyq · 2025-10-15

**Recommendation:** Reject

**Additional Comments:**

While I personally like the premises of the paper and the intended contribution, I am providing a "major revision" recommendation (formally reject in the form) following the reasoning below.

All of the reviewers report concerns over the submission and lean towards rejecting the paper. Even though some of their concerns may be overblown, the negative sentiment signals some problem on how the contributions are provided. Moreover, some legitimate concerns are common across reviewers, mainly that the formal analysis is not specific to the RL problem and that the empirical study is not conclusive for practicioners use.

I also partly share the latter concerns and find the framing of the paper confusing. I think that the submission will make for a more substantial contribution if developed in one of these directions: (1) A formal theoretical study of the impact of entropy regularization in general policy gradient procedures, (2) A more comprehensive empirical study of the impact of state entropy regularization on practical policy gradient algorithms, essentially extending the work of Ahmed et al., (2019) to state entropy regularization.

**Audience:**

Yes

**Audience Explanation:**

The topic considered in this submission looks relevant and interesting for the TMLR audience, both from the community of optimization and reinforcement learning.

**Claims And Evidence:**

No

**Claims Explanation:**

The paper addresses the impact of entropy-based regularization terms on policy gradient algorithms. First, the paper develop a set of criteria regarding the impact of regularization on the optimization landscape and the gradient estimate for generic gradient ascent procedures. Then, it provides an illustrative empirical analysis of REINFORCE with entropy-based regularization (entropy of the action distribution or the state distribution) with the lenses of the introduced criteria. The paper consider toy domains and Minigrid environments.

There is no claim that is clearly not supported, but there might be partial overclaiming. Mostly, with the introduction of the formal criteria one may expect a formal analysis of the impact of the considered regularization terms, which is instead limited to an empirical study.

**Resubmission Of Major Revision:**

The authors may consider submitting a major revision at a later time.